Violent deaths of media workers associated with conflict in Iraq, 2003–2012

Collinson Lucie lucie.collinson@otago.ac.nz
Wilson Nick
Thomson George
Department of Public Health, University of Otago Wellington , Wellington , New Zealand
Newell Marie-Louise
Electronic publication date: 2014 May 15
Publication date: 2014
Volume: 2
Electronic Location ID: e390
Received 2014 Jan 23; Accepted 2014 May 1
Copyright: © 2014 Collinson et al.
Copyright year: 2014
Copyright holder: Collinson et al.
License: This is an open access article distributed under the terms of the Creative Commons Attribution License, which permits unrestricted use, distribution, reproduction and adaptation in any medium and for any purpose provided that it is properly attributed. For attribution, the original author(s), title, publication source (PeerJ) and either DOI or URL of the article must be cited.
License URL: https://creativecommons.org/licenses/by/4.0/

Keywords: Media worker, Iraq, Epidemiology, Surveillance, Violent death

Funding: This study had no funding support.

==============================
Background. The violent deaths of media workers is a critical issue worldwide, especially in areas of political and social instability. Such deaths can be a particular concern as they may undermine the development and functioning of an open and democratic society.

Method. Data on the violent deaths of media workers in Iraq for ten years (2003–2012) were systematically collated from five international databases. Analyses included time trends, weapons involved, nationality of the deceased, outcome for perpetrators and location of death.

Results. During this ten-year period, there were 199 violent deaths of media workers in Iraq. The annual number increased substantially after the invasion in 2003 (peaking at n = 47 in 2007) and then declined (n = 5 in 2012). The peak years (2006–2007) for these deaths matched the peak years for estimated violent deaths among civilians. Most of the media worker deaths (85%) were Iraqi nationals. Some were killed whilst on assignment in the field (39%) and 28% involved a preceding threat. Common perpetrators of the violence were: political groups (45%), and coalition forces (9%), but the source of the violence was often unknown (29%). None of the perpetrators have subsequently been prosecuted (as of April 2014). For each violent death of a media worker, an average of 3.1 other people were also killed in the same attack (range 0–100 other deaths).

Discussion. This analysis highlights the high number of homicides of media workers in Iraq in this conflict period, in addition to the apparently total level of impunity. One of the potential solutions may be establishing a functioning legal system that apprehends offenders and puts them on trial. The relatively high quality of data on violent deaths in this occupational group, suggests that it could act as one sentinel population within a broader surveillance system of societal violence in conflict zones.

Introduction

Violence is a preventable cause of injuries and deaths as well as being a threat to human security (Owen, 2004). It is also a recognised dimension for measuring state governance (Kaufmann, Kraay & Mastruzzi, 2011). Data on violent deaths can refer to either interpersonal or collective violence (Dahlberg & Krug, 2002; Zwi, Garfield & Loretti, 2002), and may be an indicator of the level of conflict at a societal level.

More specifically, the violent deaths of media workers is a major issue worldwide, with persistent high levels of impunity—meaning the perpetrators are rarely prosecuted (Riddick et al., 2008). Furthermore, media worker freedom and safety is of particular concern in settings of political and social instability (Krueger, 2008). Media workers may be targeted because of their work by direct acts of violence and threats of violence during times of social and political breakdown and war, or be harmed as bystanders from more general violence.

The violent deaths of media workers can be studied in the Iraq, a setting with high levels of societal violence. The invasion of Iraq in 2003 was associated with a notable increase in the number of media workers suffering violent deaths (Wilson & Thomson, 2007) as well as an increase in civilian deaths (Roberts et al., 2004). During the years 2003–2008, it was the country with the highest number of media workers killed, and for the period 2006–2007 it contributed between 36% and 54% of media workers deaths worldwide (CPJ, 2012; IPI, 2012; RSF, 2012). Iraq continues to score highest on the Committee to Protect Journalists’ (CPJ) impunity index each year since it was first published in 2008 (CPJ, 2014). This index ranks countries based on the proportion of violent deaths of media workers where perpetrators are not prosecuted.

Given this background, we aimed to study the violent deaths of media workers in Iraq, determine the scope of ways to prevent this problem, and evaluate whether deaths in this occupational group could be used as a sentinel surveillance of societal violence in conflict zones. This latter issue is particularly relevant to Iraq, where there have been large variations in estimates of the scale of civilian deaths by various studies and surveillance systems data (Burkle & Garfield, 2013; Burnham et al., 2006; Roberts et al., 2004; Hicks et al., 2011a; Hicks et al., 2011b; Hicks et al., 2009; Hagopian et al., 2013).

Methods

Definitions

As per a previous study in which two of the authors were involved (NW and GT) (Riddick et al., 2008), we defined media workers as being those who collect or present information for public use (e.g., media presenters and translators), and those who make decisions about what information is collected and presented to the public (e.g., editors). A number of related occupations were excluded, including drivers and security guards associated with media companies. Violent death was defined as intentional violence or as a result of being in the vicinity of fighting (e.g., death from cross-fire or air strikes by military forces). To ensure a conservative estimate, we excluded violent deaths where there was insufficient evidence from the databases that the death was in anyway related to the person being a media worker or conducting their job as a media worker, for example if the body was found in a morgue and the circumstances of death were unclear (Table 1). Also excluded were suicide of the media worker and cases where a media worker was missing and presumed dead, but where a body was never found.

Table 1 Violent deaths of media workers in Iraq excluded from this analysis (2003–2012).

In the five databases reviewed, there were an additional 107 violent deaths of media workers that were excluded from our analysis. The major reasons for these exclusions were: (i) for 73, the media worker death only being recorded in one of the five databases (68%); (ii) for five, the individual not being definitively identified as being a media worker (5%); (iii) for 16, not having any names identified (15%); or (iv) for eight, only having one name identified (8%).

Reason for exclusion	N	%	
Recorded in only one of the five databases	73*	68.2	
Suspected media worker but not fully identified or name not revealed	16**	15.0	
Only first name or surname identified	8***	7.5	
Not within our definition of a media worker occupation (see Method)	5	4.7	
Insufficient evidence that the death of the media worker was in anyway related to being a media worker or conducting their job as a media worker (e.g., may have been other criminal activity such as robbery)	4	3.7	
No body recovered or no definitive proof of death	1	0.9	
Total	107	100.0	
Notes.

Data sources: Data were collected for the ten-year period 2003–2012, from five online databases: Committee to Protect Journalists (CPJ), Reporters without Borders (RSF-Reporters Sans Frontières), United Nations Educational, Scientific and Cultural Organisation (UNESCO), the International News Safety Institute (INSI) and the International Press Institute (IPI).

* RSF n = 23, CPJ n = 20, INSI n = 16, IPI n = 7, UNESCO n = 7.

** INSI n = 12, CPJ n = 4.

*** CPJ n = 4, INSI n = 4.

Data collection

Data were collected for the ten-year period 2003–2012, from five databases used in a previous study (Riddick et al., 2008) that were compiled by: the Committee to Protect Journalists (CPJ) (CPJ, 2012), Reporters Without Borders (RSF-Reporters Sans Frontières) (RSF, 2012), United Nations Educational, Scientific and Cultural Organisation (UNESCO) (UNESCO, 2012), the International News Safety Institute (INSI) (INSI, 2012) and the International Press Institute (IPI) (IPI, 2012).

Violent deaths of named individuals were only included where they were documented in two or more of the five databases. Details for each death were collected from each of the databases. Where necessary an internet search (using Google) was performed to obtain further information. Included and excluded cases were reviewed at the end of the data collection by two of the authors, to ensure that criteria had been applied consistently. Where specific nationality was not identified in the searchers, media workers were considered to be nationals of Iraq if there was evidence of long-term residence, local extended families, or local hometown provided by the source organisations. Where sex was unknown (n = 21), names were checked with a fluent Arabic speaker (whose first language was Arabic).

Media workers were often reported as having more than one occupation within the domain of media work. We recorded the occupation they were doing at the time of death, or, if it was not clear, we used the occupation they were documented as working at most commonly. The perpetrators of the violence from which the media worker died were recorded where these data were available from the source organisations, and whether or not they had been brought to justice. Data on perpetrators were predominantly only available from CPJ and were collected as coded by CPJ. Political groups included non-governmental militias and non-specified ‘armed men’ or ‘insurgents’. Iraqi Government forces are coded as militia.

Data were collected on the number of others killed or injured in the same attack as each violent death of a media worker from each of the five databases. Where there was more than one media worker killed in the same incident, the numbers of others killed and injured alongside were only counted once.

Data analysis

Basic descriptive analysis was performed along with time trends. The ten year time period was separated into two five-year periods; firstly 2003–2007 from the year of the United States (US) led invasion up to and including the “surge” of coalition military activity in Baghdad when 30,000 additional US troops entered Iraq. The second period was 2008–2012 and covered the growing autonomy of the Iraqi Government and institutions. It included the final year of coalition troop withdrawal (2011).

We used a Poisson regression in the statistical software package Stata (v 11) to calculate the confidence intervals for the ratio of counts of civilian to media worker deaths by year (thus Poisson variation is considered for the ratio of counts between these groups, rather than the ratio of the rates of events, since we had no denominator).

To provide additional context, the BBC news agency’s timeline for chronology of key events in Iraq, along with data from the New York Times, were used to inform the interpretation of the time series analysis (BBC, 2012; Livingston, O’Hanlon & Unikewicz, 2011).

Results

This analysis identified 199 violent deaths of media workers in Iraq during 2003–2012 that met our definitions. This compares with 231 deaths recorded by CPJ (including both confirmed and unconfirmed judgements of whether the death was related to the media worker’s work), 190 by RSF, 190 by IPI, 158 by INSI and 116 by UNESCO.

In the five databases reviewed, there were an additional 107 deaths of media workers that were excluded from our analysis (Table 1). The major reasons for these exclusions were: (i) for 73, the media worker death only being recorded in one of the five databases (68%); (ii) for five, the individual not being definitively identified as being a media worker (5%); (iii) for 16, not having any names identified (15%); or (iv) for eight, only having one name identified (8%).

Annual trends

The annual number of violent deaths in media workers rose from 15 in 2003 to 47 in 2007 (the peak year) dropping back to 5 in 2012 (Fig. 1). The peak years (2006–2007) for these deaths matched the peak years for estimated civilian fatalities (Fig. 3). There were no media worker deaths recorded for Iraq in 2002 in a previous study (Wilson & Thomson, 2007), and in our further examination of the databases collecting data at this time.

Figure 1 Annual trends in nationality of employer of media workers killed (Iraq 2003–2012).

The annual number of media worker violent deaths rose from 15 in 2003 to 47 in 2007 (the peak year) dropping back to 5 in 2012. The majority worked for Iraqi media agencies with this proportion increasing over time. Data sources: Data were collected for the ten-year period 2003–2012, from five online databases: Committee to Protect Journalists (CPJ), Reporters without Borders (RSF-Reporters Sans Frontières), United Nations Educational, Scientific and Cultural Organisation (UNESCO), the International News Safety Institute (INSI) and the International Press Institute (IPI).

The media workers killed were more likely to be Iraqi (85%, n = 169/199) than foreign nationals (Table 2). Of those where the foreign nationality was known, most (57%, 8/14) were from OECD countries (Table 2).

Table 2 Details collected on violent deaths of media workers in Iraq (2003–2012).

The main associations and risk factors for the violent deaths of media workers in Iraq are presented for the ten-year time period (2003–2012) and for the two five-year time periods (2003–2007 and 2008–2012).

Characteristic of the media worker
killed or event causing death	All	2003–2007	2008–2012	
	N	%	N	%	N	%	
Sex	
Male	184	92.5	148	92.5	36	92.3	
Female	15	7.5	12	7.5	3	7.7	
Age (years)	
Mean	36.5	37.0	34.8	
Median	35.0	35.0	30.0	
Range	18–75	22–67	18–75	
Nationality	
Iraqi	169	84.9	132	82.5	37	94.9	
Non-Iraqi	30	15.1	28	17.5	2	5.1	
Nationality of media worker if not Iraqi (where known)	
OECDa	8	57.1	8	66.7	0	0	
Otherb	6	42.9	4	33.3	2	100.0	
Employer nationality (country details)	
Iraq	124	62.3	92	57.5	32	82.1	
OECDc	49	24.6	47	29.4	2	5.1	
Middle Eastern Countryd	26	13.1	21	13.1	5	12.8	
Specific media occupation	
Media presentere	100	50.3	74	46.3	26	66.7	
Camera operator	38	19.1	30	18.8	8	20.5	
Editor/Deputy editor	28	14.1	24	15.0	4	10.3	
Producer	8	4.0	8	5.0	0	0	
Photographer	7	3.5	7	4.4	0	0	
Director/Head	5	2.5	4	2.5	1	2.6	
Interpreter/Translator	6	3.0	6	3.8	0	0	
Sound operator	5	2.5	5	3.1	0	0	
Otherf	2	1.0	2	1.3	0	0	
Medium worked in	
Television	103	51.8	78	48.7	25	64.7	
Print media	82	41.2	72	44.9	10	25.6	
Radio	7	3.5	6	3.8	1	2.6	
Online news agency	7	3.5	4	2.5	3	7.7	
Means of death	
Gunfire	135	67.8	111	69.4	24	61.5	
Bomb (suicide)g	15	7.5	11	6.9	4	10.3	
Bomb (non-suicide, excluding vehicle bombs)	12	6.0	10	6.3	2	5.1	
Grenade, missile or landmine	9	4.5	9	5.6	0	0	
Vehicle bomb	9	4.5	1	0.6	8	20.5	
Airstrike	6	3.0	6	3.8	0	0	
Otherh	13	6.5	12	7.5	1	2.6	
Location of the death	
On assignment in the field	77	38.7	63	39.4	14	35.9	
Travelling (other than on assignment)	47	23.6	38	23.8	9	23.1	
Home	23	11.6	16	10.0	7	17.9	
Unknown	21	10.6	17	10.6	4	10.3	
Office	10	5.0	10	6.3	0	0	
Shop or market	7	3.5	4	2.5	3	7.7	
Drive by shooting	4	2.0	4	2.5	0	0	
Otheri	10	5.0	8	5.0	2	5.1	
Threats (preceding the violent death and to family or employers following it)	
None	100	50.3	94	58.8	6	15.4	
Unknown	43	21.6	18	11.3	25	64.1	
Self/family	34	17.1	26	16.3	8	20. 5	
Employer	14	7.0	14	8. 8	0	0	
Both (self/family and employer)	8	4.0	8	5. 0	0	0	
Taken captive	
Unknown	110	55.3	84	52.5	26	66.7	
No	64	32.2	56	35.0	8	20.5	
Yes	25	12.6	20	12.5	5	12.8	
Tortured	
Unknown	111	55.8	85	53.1	26	66.7	
No	79	39.7	67	41.9	12	30.8	
Yes	9	4.5	8	5.0	1	2.6	
Perpetrators of the violence	
Political group	89	44.7	75	46.9	14	35.9	
Unknown	58	29.1	43	26.9	15	38.5	
Coalition forces	18	9.0	17	10.6	1	2.6	
Militia	15	7.5	13	8.1	2	5.1	
Suicide attacker	15	7.5	11	6.9	4	10.3	
Otherj	4	2.0	1	0.6	3	7.7	
Notes.

Data source: Data on the characteristics of the media worker killed or event causing death were collected for the ten-year period 2003–2012, from five online databases: Committee to Protect Journalists (CPJ), Reporters without Borders (RSF-Reporters Sans Frontières), United Nations Educational, Scientific and Cultural Organisation (UNESCO), the International News Safety Institute (INSI) and the International Press Institute (IPI).

a OECD, Organisation for Economic Co-operation and Development. United Kingdom (n = 2), Japan (n = 2), United States of America (n = 1), Germany (n = 1), Italy (n = 1), Poland (n = 1).

b Iran (n = 3), Palestine (n = 1), Russia (n = 1), Algeria (n = 1).

c OECD, Organisation for Economic Co-operation and Development. United States of America (n = 19), United Kingdom (n = 17), Japan (n = 3), Germany (n = 3), Spain (n = 2), Poland (n = 2), Australia (n = 1), Czech Republic (n = 1), Italy (n = 1),

d Saudi Arabia (n = 5), Qatar (n = 4), Egypt (n = 3), Iran (n = 3), Lebanon (n = 3), United Arab Emirates (n = 3), Kuwait (n = 2), Palestine (n = 2), Dubai (n = 1).

e Reporter, correspondent, broadcaster or news presenter.

f Academic (n = 1), owner (n = 1)

g One suicide bomb was also a vehicle bomb (but categorised here as suicide bomb).

h Throat slit (n = 3), beheading (n = 1), assault (n = 1), torched (n = 1), unknown (n = 7).

i Military base (n = 2), mosque (n = 2), university (n = 2), hotel (n = 1), internet café (n = 1), outside relative’s house (n = 1), walking in the centre of town (n = 1).

j Government officials (n = 2), military intelligence agents (n = 1), criminals (n = 1).

A majority (62%) of those dying worked for Iraqi media agencies. This proportion increased over time, relative to the first five-year time period but not at a statistically significant level (p = 0.053) (Fig. 1). Out of the remaining 38% not working for Iraqi media agencies, 65% worked for employers from OECD countries and 35% for employers from other Middle Eastern countries. Of the OECD countries, the USA and UK were most highly represented at 39% and 35% respectively (Table 2).

Main associations and risk factors

The major direct cause of these violent deaths was gunfire (68%), followed by suicide bombs (8%) and non-suicide bombs (6%) (Table 2). Gunfire remained the leading cause of death across both time periods with minimal variation in proportions. Deaths from grenades, missiles, landmines or airstrikes only occurred in the earlier time period (2003–07), whilst the proportion of vehicle bombs as a cause of death increased significantly in the latter time period, relative to the first five-year time period (p < 0.001).

It was difficult to classify the extent to which the media workers were intentionally sought out and killed in highly targeted attacks, versus being killed when working in the field (e.g., in a bomb blast or in cross-fire). Nevertheless, some suggestion comes from the location data in Table 2. It shows that 39% of media workers were killed whilst on assignment in the field, but most (50%) died in other settings such as whilst travelling (other than on assignment) (24%), or at home (12%) (often in front of family members). Furthermore, some of the media workers killed were reported as being tortured (5%) and taken captive (13%) prior to death, although this was often unknown (56% and 55% respectively). Media workers, their families or their employers received threats to their safety in 28% of cases.

Half of the media workers who died were media presenters (reporters, correspondents, broadcasters or news presenters) (50%), followed by camera operators (19%), and editors (14%). The proportion of media workers killed who were presenters significantly increased in the latter five-year time period relative to the first (p = 0.02).

Perpetrators and justice

Almost one-third (29%) of the perpetrators of the violent act in which the media worker died were unknown, while almost half (45%) were political groups. For all the violent deaths during the 10-year period, there has been no evidence of a consequent prosecution of perpetrators by legal authorities (as of April 2014) (CPJ, 2014). For one attack where three media workers were killed, a group of nine men were detained by the police as suspects but no further investigation was reported.

Others killed or injured alongside the media worker

Over the 10-year period, a total of 511 civilians were reported to have been killed in the same attack in which a media worker died. Furthermore, another 426 were injured in these attacks (Table 4 and Fig. 2). For each media worker killed, 3.1 civilians were killed on average in the same attack (range: 0–100) and a further 2.6 civilians were injured (range: 0–180).

Figure 2 Annual trends in number of others killed and injured in the same attacks in which media workers died (Iraq 2003–2012).

Over the 10-year period, a total of 511 civilians were reported as killed in the same attack in which a media worker died. Furthermore, another 426 were injured in these attacks. The peak year of media worker violent deaths in 2007 matched the peak year of the number of others killed and injured in the same attacks. Data sources: Data on media workers killed, others killed alongside and others injured alongside were collected for the ten-year period 2003–2012, from five online databases: Committee to Protect Journalists (CPJ), Reporters without Borders (RSF-Reporters Sans Frontières), United Nations Educational, Scientific and Cultural Organisation (UNESCO), the International News Safety Institute (INSI) and the International Press Institute (IPI).

Figure 3 Civilian and media worker deaths from violence (Iraq 2003–2012).

The peak years (2006–2007) for media worker deaths from violence matched the peak years for estimated civilian fatalities (using counts of civilian deaths from violence from the Iraq Body Count). Note: No media worker deaths from violence were recorded in Iraq in 2002 and no civilian deaths from violence were recorded by the Iraq Body Count in 2002. Thirteen media worker violent deaths were recorded for Iraq in 2013 in more than one of the five databases reviewed.

An example of two violent deaths of media workers where 10 other civilians were killed in the same attack is when Namir Noor-Eldeen and Saeed Chmagh were killed by airstrike from coalition forces whilst on assignment in New Baghdad on July 12 2007. A video of this incident is available on YouTube: https://www.youtube.com/watch?v=5rXPrfnU3G0.

The ratio of civilian to media worker deaths

Using counts of civilian deaths from violence from the Iraq Body Count (http://www.iraqbodycount.org/), the number of civilian deaths per media worker death increased from 412 (95% CI [284–597]) in 2004 to 1276 (95% CI [479–3400]) in 2009 where it peaked (Table 3).

Table 3 Ratio of violent civilian deaths to violent media worker deaths.

The number of violent civilian deaths per violent media worker deaths increased from 412 (95% CI [284–597]) in 2004 to 1276 (95% CI [479–3400]) in 2009 where it peaked.

Year	Civilian deaths (n)	Media worker deaths (n)	Ratio of civilian to
media worker deaths	95% confidence intervals
of ratio of civilian to
media worker deaths	
2003	12,093	15	806	(486–1338)	
2004	11,540	28	412	(284–597)	
2005	16,161	28	577	(398–836)	
2006	29,054	42	692	(511–936)	
2007	25,316	47	539	(405–717)	
2008	9639	14	689	(408–1163)	
2009	5102	4	1276	(479–3400)	
2010	4109	7	587	(280–1232)	
2011	4147	9	461	(240–886)	
2012	4573	5	915	(381–2198)	
2003–2007	94,164	160	589	(504–687)	
2008–2012	27,570	39	707	(516–968)	
Notes.

Data sources: Civilian death counts were collected from the Iraq Body Count (Iraq Body Count, 2013). Media worker death counts were collected for the ten-year period 2003–2012, from five online databases: Committee to Protect Journalists (CPJ), Reporters without Borders (RSF-Reporters Sans Frontières), United Nations Educational, Scientific and Cultural Organisation (UNESCO), the International News Safety Institute (INSI) and the International Press Institute (IPI).

Table 4 Timeline of major events and violent deaths of media workers and others killed or injured in the same attack and deaths of other populations in Iraq (2003–2012).

Over the ten-year period, a total of 511 civilians were reported as killed in the same attacks in which a media worker died, and another 426 were non-fatally injured. The peak years of media worker violent deaths coincide with the years of the highest levels of violence in Iraq (2006 and 2007) when explosive incidents were at their height.

Year	Major socio-political events
and selected economic indicators	Violent deaths of
media workers (N)	Others killed in the
same attack killing
a media worker (N)	Others injured (non-fatally)
in the same attack (N)	Civilian deaths from
violence (N)	Homicides of Iraqi
military and police (N)	
2003	US-led invasion begins on 19 March. By 9 April much of Baghdad is under US control. On 1 May US President Bush declares military phase in Iraq has ended. On 19 August, suicide bomber bombs UN headquarters in Baghdad. On 13 December the former ruler of Iraq, Saddam Hussein, is captured by US soldiers. Post-invasion, electricity production in Iraq is reported to be: 3,500 megawatts, GDP: US $14 billion, internet users: 5000, telephone subscribers: 600,000.	15	29	23	12,093	1300	
2004	On 25 January a former CIA inspector states that Iraq has no illegal weapons. In April–May there is a month long US military siege of Falluja. On 28 June US transfers formal sovereignty of Iraq to its new leaders.	28	126	17	11,540		
2005	Surge in vehicle bombings, bomb explosions and shootings.	28	23	6	16,161	2545	
2006	Saddam Hussein tried and executed by Iraqi authorities	42	43	13	29,054	2091	
2007	A “surge” of 30,000 additional US troops enter Iraq. Improvements are seen with electricity production: 4000 megawatts, GDP: US $32 billion, internet users: 500,000, and telephone subscribers: 10 million	47	150	194	25,316	1830	
2008	Iraqi prime minister orders crackdown on militia in Basra.	14	4	4	9639	1070	
2009	Iraq takes control of security in Baghdad’s fortified Green Zone and assumes more powers over foreign troops. Al-Qaeda linked group “Islamic State of Iraq” claim responsibility for a wave of suicide bombings in Baghdad.	4	30	6	5102	515	
2010	Parliamentary elections are held. The last US combat brigade leaves Iraq.	7	13	40	4109	468	
2011	Nationwide protests calling for reform and end to corruption. The remaining US troops leave in December. Further improvements are seen with electricity production: 6500 megawatts, GDP: US $108 billion, internet users: 2 million, telephone subscribers: 22 million.	9	93	123	4147	Not reported	
2012	Bombings and gun attacks target Shia areas	5	0	8	4573	Not reported	
Total		199	511	426	121,734	9819	
Notes.

Peak numbers of deaths in each category is bolded. The Iraq body Count includes police deaths among its’ civilian death counts. Data sources: Major socio-political events were collected from the British Broadcasting Corporation (BBC, 2012). Selected economic indicators were collected from Livingston, O’Hanlon & Unikewicz (2011). Violent deaths of media workers and others killed and injured in the same attacks were collected for the ten-year period 2003–2012, from five online databases: Committee to Protect Journalists (CPJ), Reporters without Borders (RSF-Reporters Sans Frontières), United Nations Educational, Scientific and Cultural Organisation (UNESCO), the International News Safety Institute (INSI) and the International Press Institute (IPI). Civilian death counts were collected from Iraq Body Count (2013). Homicides of Iraqi military and police were collected from O’Hanlon & Livingston (2011).

Relating annual trends to key socio-political events

The annual number of violent deaths of media workers increased substantially after the military invasion in 2003 (it was zero in 2002, see above). The peak years of these deaths coincided with the years of the highest levels of violence in Iraq (2006 and 2007) when explosive incidents were at their height (Table 4). The peak year of these deaths (and others injured alongside each death) also coincided with the military surge with an additional 30,000 US troops entering Iraq. The number of violent deaths of media workers then declined in the subsequent period where the war changed from US vs Iraqi to a largely civil war with the departure of coalition forces from 2009 onwards.

Discussion

Main findings

This analysis confirms the substantial size of the problem of violent deaths of media workers in Iraq. Not only were 199 media workers killed (between 2003 and 2012), but there were a further 511 civilians killed and 426 injured in the same attacks. This picture adds evidence for the high risk of the media worker occupation in politically unstable states (Riddick et al., 2008).

These data also indicate a marked increase in the numbers of media workers dying violently from the time of the invasion of Iraq (2003) and a decrease at the time that coalition forces withdrew. The peak years (2006–2007) for violent deaths in media workers matched the peak years for estimated civilian fatalities, with similar time trends overall.

There are variable levels of evidence around the extent of intentional targeting of these media workers because of their occupation, but in some cases it is fairly suggestive e.g., being killed in their office at work and having had preceding threats.

The risks for media workers may be heightened by the apparent high levels of impunity for their killers (Khan, 2007). Indeed, in our study we found no evidence that any of perpetrators of the violent deaths of media workers recorded in Iraq during this period have been prosecuted by legal authorities. Improving the safety for media workers is likely to be imperative for the existence of a free press which is a fundamental constituent to a functioning democracy. Journalism is a vital component to emerging democratic societies (McNair, 2009) and a free press is associated with lower levels of corruption (Brunetti & Weder, 2003). A restricted press not only precludes the human right of freedom of information, but may also enable other violations of human rights and other injustices to coexist. The source organisations CPJ and IPI call for proper enforcement of the law to protect journalists whilst RSF contests laws that restrict freedom of information. INSI urges for safer working in hostile areas and provides free safety training courses for journalists worldwide. UNESCO condemns the killing of journalists and hosts World Press Freedom Day.

The similarity in trends of violent deaths of civilians and media workers suggests that deaths in this occupational group could potentially contribute as a sentinel surveillance system. For example, as one component of a broader surveillance system for monitoring societal violence in conflict zones. A particular advantage of this occupational group for this role would be the relatively reliable data on each violent death. Also, these deaths are likely to highlight the difficulties in achieving non-corrupt and democratic institutions in the conflict area. The disadvantages are that these deaths may not be fully representative of trends in other civilian deaths. This is because of the unique aspects of the occupation (e.g., travel to conflict areas) but also because media workers may be more readily able to modify their behaviour over time to reduce risk compared to other citizens (e.g., wearing body armour and hiring protection). The crude nature of only using data from media workers for sentinel surveillance is also suggested in our data where the ratio of civilian deaths to media workers deaths fluctuated markedly over the 10 year period (Table 3), as did the range of others killed or injured alongside the media workers (ranging from 0 to 100).

Study limitations

The study was dependent on the sensitivity and the accuracy of the five main databases used in documenting violent deaths of media workers. Overall the CPJ database provided more information on each of the variables collected than the other four databases. Often the data from the other databases was very similar and sometimes had identical wording. The CPJ provided most of the information on impunity, perpetrators and whether the media worker was tortured, taken captive or threatened prior to being killed. The coding of perpetrators collected from the CPJ database appears to be relatively simplistic. It does not take into account multiple perpetrators, such as the occasional example of coalition forces backing Iraqi Government forces. However, validity checks using the search engine Google did not indicate that there was any significant concern with the accuracy of coding in the CPJ database.

Four of the databases we used (CPJ, RSF, IPI, INSI) are compiled by international non-governmental organisations, and one is an official international organisation (UNESCO) with an interest in communication, media freedom, or media safety. CPJ, RSF, and IPI all have local employees or volunteers who do independent research, fact-finding missions and talk to people in the field (often to other media workers). The latter three organisations were also honoured by the U.S. National Television Academy at the News and Documentary Emmy awards in 2006. INSI do not document how it monitors media worker casualties, but they do however have links to the International Federation of Journalists. UNESCO also does not document how their list of media workers killed is compiled.

Factors that may increase the likelihood of violent deaths of media workers also increase the difficulties in gathering data on such deaths. Therefore it is possible that there is under-recording of these deaths, and that some of the cases that we excluded (n = 107) are indeed of media work-related deaths. Because of our inclusion criteria, requiring the violent death to be recorded in more than one database, and excluding those where there was insufficient evidence to suggest that the death was work related, the total is likely to be conservative.

In this study there were inadequate denominator data to allow calculations of the rate of violent death per 1000 media workers in Iraq. This problem of inadequate denominator data has been reported in other such studies (Taback & Coupland, 2006), and indeed developing a register of media workers in a country like Iraq could pose actual risks to such workers.

A further limitation is that our source for civilian deaths, the Iraq Body Count, is likely to be underrepresenting death rates (Carpenter, Fuller & Roberts, 2013; Hagopian et al., 2013). In addition the Iraq Body Count data includes media worker deaths among civilian deaths and these were not removed from the civilian death counts. Therefore the true ratios of violent civilian deaths to violent deaths of media workers would be slightly lower than the ratios presented here, as some of the media worker deaths in the numerator (data collected from the five databases) would also be in the denominator (Iraq Body Count data) (Table 3).

Implications for further research

Further studies are needed to obtain a better understanding of the epidemiology of the violent deaths of media workers in Iraq and other conflict zones. In particular, the priority for research in such settings would be to establish plausible denominator populations of media workers—ideally both indigenous and those visiting on assignment from other countries.

International organisations could also potentially improve collaboration to limit the number of databases collecting information on these deaths, while establishing systems that collect more in-depth information (for example to better clarify if the media worker was specifically targeted and if perpetrators were processed by the justice system).

Strengthening data collection on violence has been nominated as a goal by the WHO Global Campaign for Violence Prevention 2012–2020 (Butchart, Mikton & Kieselbach, 2012). Given the lack of reliability of government reports related to civilian deaths, where gaps in mortality and morbidity data that relate to violence exist, there is a need to coordinate what resources are currently available and utilise them. In some cases there is a need to establish independent surveillance systems of violent death in conflict zones.

Conclusions

This paper highlights the high number of violent deaths of media workers in Iraq in this conflict period, in addition to the high levels of impunity for perpetrators. Many others also die and are injured when a media worker is killed in an attack. This situation suggests a need for urgent preventative measures to protect the safety of media workers (for example by having an effective policing and legal system). Collecting data on media workers could also potentially be a sentinel surveillance system that contributes to a broader surveillance system of societal violence in conflict zones.

We thank those organisations that have collected and made publicly available data on violent deaths of media workers (see Methods). We thank Dr James Stanley for his assistance with some of the statistical analysis and Dr Sarah Al Youha for her assistance with identifying the gender of the first names of media workers where it was not recorded.

Additional Information and Declarations

Competing Interests

Author Contributions

The authors declare there are no competing interests.

Lucie Collinson performed the experiments, analyzed the data, contributed reagents/materials/analysis tools, wrote the paper, prepared figures and/or tables, reviewed drafts of the paper.

Nick Wilson conceived and designed the experiments, analyzed the data, contributed reagents/materials/analysis tools, wrote the paper, prepared figures and/or tables, reviewed drafts of the paper, Nick Wilson oversaw all aspects of the study.

George Thomson conceived and designed the experiments, contributed reagents/materials/analysis tools, wrote the paper, reviewed drafts of the paper, George Thomson contributed to the data interpretation.

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
