# Peer review of "Violent deaths of media workers associated with conflict in Iraq, 2003–2012"

_PeerJ, doi:10.7717/peerj.390_

## Round 0.1 · original submission · Major Revisions

The paper is of interest, but as it stands it is essentially a report of cases, and it is difficult to interpret this at a population level. The reviewers make many good suggestions to improve the paper, to which I would add:

Introduction: paragraph 3, clarify whether these media workers are targetted specificically because they are media or because they happen to be there?

paragraph 4 of the introduction: was the increase in the number of media workers deaths in 2003 worse than among civilians?

In an epidemiological study one would not only present the cases, but also present the denominators, in this case of media workers, and ideally there should also be an group of non-media workers for comparison.

Page 3 top - if workers are targetting in the office how relevant is it that they are media workers or other workers?

Page 3, third paragraph: avoid the word gender here, and use sex, as gender has socio-cultural meanings.

page 4, top: where does the count for civilian deaths come from?

page 4, annual trends - do you know the numbers of media workers in the country per year?

·

Basic reporting

Please see attached letter.

Experimental design

Please see attached letter.

Validity of the findings

Please see attached letter.

Reviewer 2 ·

Basic reporting

1. In all cases, the authors still need to integrate their references into all appropriate points in the text of their paper. As it is, it is impossible to evaluate the quality of this paper to a meaningful degree, or the authors’ use of the literature, because the references (listed at the end of the paper) are not cited within the paper itself. The Introduction, Methods and Discussion are replete with statements that are currently unsubstantiated by any citation. The paper’s format is not currently in keeping with any academic format and this needs to be revised in order to review the paper.

2. RE “Others killed or injured alongside the media worker”: The authors’ Figure 2 cited here does not match the actual Figure 2. Please correct your Figures/Figure citations carefully.

3. RE Table 2:
a. Why are there numbers to the right that match nothing in the blank row between ‘male’ and ‘female’?
b. You have the same problem with numbers elsewhere in the table that align with a blank or missing heading on the left. Please correct.
c. Please spell out words for the acronym “OECD” in its footnote 1.
d. The heading “Degree of intentionality of the homicide in relation to media work”, with “killed on assignment” beneath it illustrates the confusion over how authors defined “intentionality”, which I describe more in comments later: Being killed by a gun, mortar, or air strike while working on assignment in the middle of a battle zone does not inherently mean that the individual was killed intentionally, or intentionally knowing he/she was a civilian, or intentionally knowing he/she was a media worker. For this reason, I would have preferred ideally an analysis that included all deaths of media workers by armed violence, with a subgroup that is definitely‘targeted/i.e. intentionally killed’ as a civilian, or as a media worker’, rather than calling all of these deaths ‘intentional’. Although I would not ask the authors to change their primary outcome to 'killed by armed violence' or 'violent deaths', I think that the authors do need to define more clearly, and describe more clearly, how they applied 'intentional'. If they cannot operationalize the outcome of 'intentional' deaths in a clear, replicable, and valid way, then they will need to revise their use of these terms substantially. I comment more on this below.

4. RE Table 3: The paper needs to acknowledge that the IBC data includes media worker deaths, and that the authors did not extract these deaths from the IBC dataset so the true ratios would be slightly lower than the ratios here.

5. Figure 1: cite the data sources

6. Figure 2: cite the data sources and cite in correct location of the paper text.

7. Figure 3: The Note is confusing about whether no deaths are recorded in 2002 because the data were gathered and there were no deaths to record, or whether no data were gathered for 2002. For example, when you write that “no civilian deaths from violence were recorded by Iraq Body Count in 2002”, this is actually because IBC did not exist in 2002, and it also did not retrospectively record for 2002 – that is, IBC did not gather data in 2002 and its data begins in 2003. The current note incorrectly gives the impression that IBC was gathering data for 2002 and there were no deaths to record. Please be clear what the message is about “No media worker homicides were recorded in Iraq in 2002: Is this because no one was gathering data on this in 2002? Or was this because there were no media worker deaths to record in 2002?

Experimental design

8. Please delineate more carefully and convincingly the operationalized definitions: ‘homicide’, ‘intentional’, ‘targeted’. For example, I was confused by how the authors identified ‘air strikes by military forces’ as ‘intentionally’ killing individual media workers. If they were known to be civilians, or media workers, then this would be obviously a war crime, as civilians – including media workers - are not supposed to be knowingly targeted for harm. If the media workers were mistaken for combatants, or accidentally killed, this would still be homicide, and it might be argued that the deaths were from indiscriminate fire. The authors seem to write that they excluded ‘crossfire’ deaths from their definition of ‘homicide’, so was this an intentional coalition attack on a media center or on a media worker, knowing that the individual was a media worker? How did they determine that this was intentional killing of a media worker? Were these intentional killings, but of media workers seemingly misidentified as combatants? If so the authors should explain this. Similarly, the car bombs and suicide bombs are not clear in how they are targeting individuals as media workers. Are they intentional (i.e. homicides) on civilians, among which they are media workers incidentally? An execution is an obvious targeting of an individual, but the examples I raise require clearer explanation of how the authors applied their definitions. I would have found it clearer to be informed of deaths by media workers by ‘armed violence’, which was then broken down by subcategory of weapon type, and also by subcategory of situation such as: cross-fire/targeting of presumed combatant groups; intentional targeting of civilians generally; intentional targeting of the media worker due to job as media worker. I am not sure that using an essentially legal definition of ‘homicide’ is helpful if the authors are actually talking about intentional/targeted deaths. It is not clear why the authors use the term homicide as well as ‘intentional’ and ‘targeted’, since all homicides are intentional, and targeting and intentional are interchangeable terms.
9. The authors might see these publications on assessing intentional killing of civilians or media workers:
a. Eck K, Hultman L. One-sided violence against civilians in war: insights from new fatality data. J Peace Research. 2007;44:233–246.
b. Hicks MH, Lee UR, Sundberg R, Spagat M. (2011) Global comparison of warring groups in 2002-2007: fatalities from targeting civilians vs. fighting battles. PLoS ONE 6(9): e23976.
c. War on Words: Who Should Protect Journalists? By Joanne M. Lisosky, Jennifer Henrichsen, Jennifer R. Henrichsen (2011)
10. In addition to explicating how they operationalized their definitions more clearly, the paper would benefit from some case examples of different types of homicides of media workers.
11. RE ‘Perpetrators: Do ‘political groups’ include Iraqi government forces, non-governmental militias and criminal groups? I agree with not using the common term ‘insurgents’, but I suggest that the authors add more detail, as political groups would normally suggest legitimate political parties. Please add more detail.
12. RE “Others killed or injured alongside the media worker”: How is it happening that civilians are killed “alongside” media workers? Is this because the media workers are not actually targeted because of their job, but are in a situation where civilians are targeted? Or is it that non-media worker civilians are killed incidentally when a media worker is targeted? More detail please.
13. RE “The ratio of civilian to media worker deaths:” The Iraq Body Count data include media worker deaths among civilian deaths. This would have only a slight effect on your ratio values, but you need to acknowledge that you did not cross-check names to extract out IBC-recorded media deaths, so some of the media deaths in your numerator (your data) may also be in the denominator (IBC data).
14. RE “Relating annual trends to key socio-political events”: RE Table 4: IBC includes police deaths among its civilian death counts.
15. In your Discussion and Methods: Please describe more clearly how you count deaths of media workers that “purposefully go into hazardous areas as part of their work (e.g. where they risk being killed in cross-fire)”. So do your 199 deaths really exclude those killed by cross-fire in battle? How did you define ‘cross-fire’ – e.g. is this in an active two-sided fire-fight? And does this mean that you include among your ‘intentional deaths’, i.e. homicides, the media workers who go into battle and are killed by one side, (e.g. by an air-strike or by a mortar), assuming that they should be known to be media workers rather than combatants and that they were specifically targeted as media workers? Or is that considered an accidental death and not targeted? You could argue that when media workers are among combatants that do not wear uniforms, they are not carrying weapons, and so should not be mistaken for, or assumed to be, combatants even when they are among un-uniformed combatants. In other words, you could make the case that in a conflict such as Iraq where some combatant groups do not identify themselves by uniforms, the onus is on the opponent to discriminate to the best of their ability civilians from un-uniformed combatants, e.g. by not killing adult males who are not carrying a weapon.

Validity of the findings

16. RE “Conclusions”: I suggest that the conclusions focus more on the study’s findings. Some of the implications of the study here could be moved into the last paragraph of your Discussion.

Additional comments

This paper is important in identifying deaths of media workers in the context of a specific armed conflict, which is an under-recognized area in the literature. I hope that the suggested revisions are helpful to improve the academic quality and clarity of the paper.

---

## Round 0.2 · Minor Revisions

I have amended the text in the word (track changes) document, to make the presentation a bit tighter. I will email this document separately to the corresponding author, who basically only needs to agree, accept the changes and resubmit a clean version of the paper.

---

## Round 0.3 · accepted · Accept

Thank you for your patience in revising the manuscript, which I hope you now feel is a much improved version.